# Combining cognitive bias modification training (CBM) and transcranial direct current stimulation (tDCS) to treat binge eating disorder: study protocol of a randomised controlled feasibility trial

Gemma Gordon  ,[1] Timo Brockmeyer,[2] Ulrike Schmidt,[1] Iain C Campbell[1]

[1]Section of Eating Disorders, Department of Psychological Medicine, Institute of Psychiatry, Psychology and Neuroscience, King's College London, London, UK
[2]Department of Clinical Psychology and Psychotherapy, Institute of Psychology, University of Gottingen, Goettingen, Germany

**Correspondence to**
Gemma Gordon;
gemma.gordon@kcl.ac.uk

## ABSTRACT

**Introduction** Binge eating disorder (BED) is a common mental disorder, closely associated with obesity. Existing treatments are only moderately effective with high relapse rates, necessitating novel interventions. This paper describes the rationale for, and protocol of, a feasibility randomised controlled trial (RCT), evaluating the combination of transcranial direct current stimulation (tDCS) and a computerised cognitive training, namely approach bias modification training (ABM), in patients with BED who are overweight or obese. The aim of this trial is to obtain information that will guide decision-making and protocol development in relation to a future large-scale RCT of combined tDCS+ABM treatment in this group of patients, and also to assess the preliminary efficacy of this intervention.

**Methods and analysis** 66 participants with Diagnostic and Statistical Manual-5 diagnosis of BED and a body mass index (BMI) of ≥25 kg/m² will be randomly allocated to one of three groups: ABM+real tDCS; ABM+sham tDCS or a wait-list control group. Participants in both intervention groups will receive six sessions of ABM+real/sham tDCS over 3 weeks; engaging in the ABM task while simultaneously receiving bilateral tDCS to the dorsolateral prefrontal cortex. ABM is based on an implicit learning paradigm in which participants are trained to enact an avoidance behaviour in response to visual food cues. Assessments will be conducted at baseline, post-treatment (3 weeks) and follow-up (7 weeks post-randomisation). Feasibility outcomes assess recruitment and retention rates, acceptability of random allocation, blinding success (allocation concealment), completion of treatment sessions and research assessments. Other outcomes include eating disorder psychopathology and related neurocognitive outcomes (ie, delay of gratification and inhibitory control), BMI, other psychopathology (ie, mood), approach bias towards food and surrogate endpoints (ie, food cue reactivity, trait food craving and food intake).

**Ethics and dissemination** This study has been approved by the North West-Liverpool East Research Ethics Committee. Results will be published in peer-reviewed journals.

**Trial registration number** ISRCTN35717198

### Strengths and limitations of this study

► The Investigating Concurrent Approach Bias Modification Training and Transcranial Direct Current Stimulation in Binge Eating Disorder (ICARUS) study is the first randomised controlled feasibility trial of multi-session transcranial direct current stimulation (tDCS) combined with cognitive bias modification training (CBM) for adults with binge eating disorder (BED).
► ICARUS will compare (tDCS+CBM) versus (sham tDCS+CBM) and a wait-list control group.
► ICARUS is designed to answer questions about the efficacy of the treatments tested.
► Results would need to be replicated in a larger trial before recommendations for tDCS+CBM as a treatment adjunct for patients receiving outpatient treatment for BED can be made.

## INTRODUCTION

Binge eating disorder (BED) is the most prevalent eating disorder (ED) worldwide, with 1%–3% of the general population meeting diagnostic criteria.[1 2] Binge eating is a core symptom, characterised by consumption of large amounts of food, a sense of loss of control and significant distress. Nearly 80% of those with lifetime BED have a comorbid psychiatric disorder, such as mood, anxiety, substance use disorders or another ED.[2] Due to the lack of compensatory behaviours (eg, vomiting, excessive exercising), BED is often accompanied by, or leads to, obesity and associated physical complications.[3 4] In the general population, approximately 30%–42% of people with BED are obese.[2 5 6] Around 30% of treatment-seeking obese people[7–9] and up to 47% of bariatric surgery candidates have full or partial BED.[1 10 11] While BED itself has considerable individual and societal costs,[12] the combination of BED and obesity

is associated with more severe obesity, greater medical and psychiatric comorbidity, greater functional impairment and perinatal complications.[12–15] Treatments for BED and obesity are sub-optimally effective, with cognitive behavioural therapy (CBT)[16] and some medications[17] reducing binge eating and related psychopathology,[1] and approximately 50%–60% of patients achieving abstinence from bingeing at the end of treatment[18] with some sustained cessation at follow-up.[19] However, drop-out rates in established BED treatments reach 12%–34%, and 30%–50% of BED patients relapse in long-term follow-ups,[20–22] indicating that a substantial proportion do not maintain binge eating remission. Lisdexamfetamine[23] and topiramate[24] also reduce weight in the short-term but have considerable side effects,[25] and their longer term efficacy is uncertain. Thus, there is a need for novel treatment developments.

The aetiology of BED is widely seen as multi-factorial. Emerging neurobiological models emphasise both the role of stress in the onset and maintenance of the disorder,[26 27] and the development of addiction-like features; craving, tolerance and binge escalation over time,[28 29] impulsivity and compulsivity, alterations in executive function and attention[30] and reward-related decision-making.[31]

On encountering images of high-calorie food, BED patients report enhanced reward sensitivity and exhibit stronger medial orbitofrontal cortex responses compared with healthy controls and participants with bulimia nervosa (BN).[32] In individuals with obesity, who may or may not have BED, activation in the ventral striatum (part of the reward system) has been found to be higher compared with normal-weight controls,[33] in tandem with a more pronounced approach bias towards appetising food images,[34 35] leading to greater likelihood of consumption. Furthermore, poor reward-related decision-making behaviour may be a maintaining factor in obesity.[36] Converging data using different methodologies, such as brain imaging, eye tracking and behavioural test paradigms[37] have found that patients with BED demonstrate a higher arousal rate in response to food stimuli, a concurrent motor plan to start eating, a higher reward sensitivity and greater inhibitory deficits as compared with individuals without BED.[32 38 39] Those with obesity and BED (compared with obesity alone) have demonstrated that their attentional bias to food images held higher motivational value,[40] and responded more to high calorie food images in sites of cognitive planning of motor movements, driven by emotions, which may reflect impulsive tendencies in the face of a binge-eating trigger. This tendency to approach and consume palatable food items may thus be compounded by a greater sensitivity to reward and a decreased capacity to inhibit action tendencies. This is corroborated by the recent finding that individuals with BED or BN show higher food cue reactivity (increased cravings) when exposed to visual food cues compared with healthy controls.[41] Such accumulating evidence of BED as a unique diagnostic group situates it as a distinct phenotype within the obesity spectrum that is characterised by increased impulsivity.[42]

Conventional treatments of BED, such as CBT may not be best suited to target highly automatic cognitive processes that occur at an early stage in information processing and that are considered to contribute to food craving and associated maladaptive cognitions/behaviours. Two 'brain-directed' treatments may provide an avenue for modifying these processes: approach bias modification training (ABM) and transcranial direct current stimulation (tDCS).

ABM is a form of cognitive bias modification training (CBM) that aims to retrain approach bias tendencies (reach out towards) into avoidance ones (move away from)[43] regarding stimuli such as appetitive cues. Participants are systematically trained to show an avoidance movement in response to illness-related rewarding stimuli (eg, food or alcohol) on a computer screen. ABM techniques have shown potential in several pilot and large-scale randomised controlled studies to treat alcohol[44] and tobacco[45] addictions, and to reduce consumption of cannabis[46] and unhealthy foods.[47 48] ABM has also yielded promising results in people with high levels of food craving and in bulimic EDs, including BED.[49 50] However, mixed results in empirical studies across these domains[51 52] raise methodological issues in ABM studies to date, such as low statistical power and suboptimal choice (or absence) of control groups[53] and administration of single versus multiple training sessions.

tDCS is a form of non-invasive brain stimulation (NIBS) that has been used as a treatment adjunct for a range of psychiatric disorders, such as depression, schizophrenia and addictions.[54–56] Preliminary evidence suggests that tDCS and other forms of NIBS are promising tools to reduce food cravings, ED symptoms and body weight in bulimic EDs, including BED, and obesity.[57] Additionally, some studies indicate that NIBS may reduce depression/stress levels and improve reward-based decision-making in ED patients.[58] A frequent stimulation target is the dorsolateral prefrontal cortex (dlPFC) which plays a major role in cognitive-inhibition, emotion regulation and reward processing.[58–61] Although precise mechanisms of action of tDCS have yet to be understood, a key hypothesis in relation to BED is that enhancing dlPFC activity via tDCS alters the reward-cognition balance towards facilitation of cognitive control and suppression of reward-related mechanisms driving food craving/overeating.[62]

If given concurrently (ie, 'online training'), NIBS is reported to boost the effects of cognitive training on the reduction of cognitive biases and the improvement of response inhibition.[63] NIBS may enhance synaptic strength in neuronal pathways activated by cognitive training, amplifying effects of training and thus cognitive bias modification efficacy.[64] As the effectiveness of tDCS may thus be improved by pairing administration with a cognitive task inducing activity in the target brain region,[65–67] such combined treatment interventions have been investigated among alcohol dependent inpatients

(ABM and tDCS),[68] and to enhance inhibitory control related to food consumption (Go/No-Go Task and tDCS).[65] The insignificant findings from these studies warrant commentary that to date, studies that have found positive effects of tDCS have either included obese participants or have had multi-session protocols.[65 69–71] As this study incorporates both aspects, it is optimally designed to yield significant results.

In light of both the individual and societal burden incurred by the rising prevalence of BED and obesity, research interventions informing treatments that lead to stable and long-lasting remission are of critical importance, and novel therapies may play a role in serving as adjuncts to treatment as usual (TAU), to enhance improvement in clinical outcomes obtained from engaging with ED treatment services. This research trial is the first to combine two promising novel intervention strategies in an integrated treatment and will yield important findings to shape future clinical trials. The intervention conditions of this feasibility study will involve six sessions of concurrent ABM and real or sham tDCS over 3 weeks, and will assess participant acceptability and dropout rates at this treatment frequency and duration. Additionally, the frequency of participants' ED symptoms and other outcomes related to general psychopathology and neuro-cognition will be measured before and after the study interventions to assess treatment success. In summary, this proof-of-concept and feasibility study will establish the utility of concurrent ABM+real tDCS in improving clinical outcomes in participants with BED, compared with ABM+sham tDCS, and a wait-list control group.

## STUDY AIMS

In line with established recommendations for outcomes of feasibility trials,[72] which at present are supported by the National Institute for Health Research, the primary aim is to assess the feasibility of using concurrent ABM+real tDCS compared with concurrent ABM+sham tDCS as a potential adjunct to TAU in this patient population, and acquire key information to inform the development of a large-scale RCT.

The specific objectives of the proposed feasibility study are to:

1. Establish the feasibility of conducting a large-scale RCT of ABM+tDCS in patients with BED by assessing recruitment, attendance and retention rates.
2. Determine the practicality of administering both ABM and tDCS simultaneously.
3. Determine the best instruments for measuring outcomes in a full trial by examining the quality, completeness and variability in the data.
4. Estimate the treatment effect sizes and SD for outcome measures to inform the sample size calculation for a large-scale RCT.
5. Evaluate whether the treatment is operating as it is designed by analysing process measures, such as with-in-session Visual Analogue Scales (VAS) of key ED symptoms.
6. Determine whether patients with BED evaluate concurrent ABM+tDCS as acceptable and credible.
7. Obtain information about patients' willingness to undergo random allocation to ABM paired with either real or sham tDCS administration, or the wait-list control condition.

A secondary aim is to investigate the potential efficacy of concurrent delivery of both forms of treatment on BED.

This will involve evaluating if:

1. Concurrent sessions of ABM+real tDCS are superior to ABM+sham tDCS and to wait-list control in terms of frequency of objective binge eating episodes, food cue reactivity, food craving, food intake, ED psychopathology and mood.
2. Concurrent ABM+tDCS is superior to the two other conditions in having an effect on the targeted neuro-cognitive mechanism (approach bias for high calorie food) and related neurocognitive parameters (ie, impulsivity, delayed gratification, emotional regulation).
3. Concurrent ABM and sham tDCS is superior to the wait-list control in eliciting therapeutic effects on the aforementioned clinical outcomes and neurocognitive mechanisms, yet demonstrates an efficacy level below that of concurrent ABM and real tDCS.

## METHODS AND ANALYSIS

This study protocol has been written according to the Standard Protocol Items for Randomised Trials statement[73] and the Consolidated Standards of Reporting Trials (CONSORT) 2010 statement.[72]

### Study design

The Investigating Concurrent Approach Bias Modification Training and Transcranial Direct Current Stimulation in Binge Eating Disorder (ICARUS) trial is an exploratory randomised controlled feasibility trial with three parallel treatment conditions; ABM+real tDCS, ABM+sham tDCS and wait-list control. All participants across the two intervention groups will receive a treatment protocol of six sessions of ABM+real/sham tDCS conducted over 3 weeks. The comparator groups of a wait-list control and ABM+sham tDCS are necessary to evaluate the potential effect of real versus sham tDCS in participants with BED. The wait-list control group will be examined at the same time points to control for the possibility that improvements in the intervention groups are simply due to regression to the mean, spontaneous remission or other non-specific time effects. Any participants who are engaging in treatment for their ED will continue with TAU, and thus this selection of comparators is deemed acceptable. Within treatment session measures will involve VAS evaluating mood, stress and

ED symptoms. Assessments will be conducted three times during the study; at baseline, post-treatment (week 3) and at follow-up (week 7).

## Participants

Inclusion criteria entail: (1) male and female community-dwelling adults (aged 18–70), (2) overweight or obese according to WHO criteria (body mass index (BMI) $\geq 25\,kg/m^2$),[74] (3) a diagnosis of full-syndrome or sub-threshold BED according to the Diagnostic and Statistical Manual (DSM)-5[75], (4) fluency in English, (5) normal or corrected to normal vision.

Exclusion criteria entail: (1) all known contraindications to tDCS[76]; (2) pregnancy; (3) a current significant/unstable medical or psychiatric disorder needing acute treatment in its own right; (4) a lifetime diagnosis of substance dependence, psychosis, bipolar disorder or borderline personality disorder; (5) taking psychotropic medication other than a stable dosage of selective serotonin reuptake inhibitors for at least 14 days prior to study enrolment; (6) allergies to any of the foods presented in the study; (7) smoking >10 cigarettes per day; (8) drinking >3–4 units (men) or 2–3 units (women) of alcohol per day. In line with the CONSORT guidelines,[77 78] we will record the number and reasons for any participants we must exclude, or any who decline consent or withdraw from the study.

## Sample size

As ICARUS is a feasibility study, an a priori sample size calculation is not necessary. Rather, its aim is to provide effect sizes on which future large-scale studies can be powered. Total study sample sizes of n=24 to n=50 have been recommended for feasibility trials with a primary outcome measured on a continuous scale, mainly because estimates of the SD for normally distributed variables tend to stabilise around this size.[79 80] We have chosen a target end study sample size of n=60, (ie, exceeds the upper end recommended for feasibility trials). However, assuming the attrition to follow-up rate is a=0.10 (as found in previous ED trials[81 82]) and applying an attrition correction factor of 1/(1-a), we will recruit an actual sample size of 66, that is, 22 participants per group.

## Randomisation

After the baseline assessment, participants will be allocated to one of three conditions at random to receive six sessions of either concurrent ABM+real tDCS or ABM+sham tDCS, or no intervention in the wait-list control condition. As a proportion of participants recruited from an outpatient ED clinic will be on a waiting list to receive treatment at the time of enrolment in this study and may commence treatment shortly after study enrolment, this study will not seek to balance groups in terms of therapy engagement or medication usage. Participants in the wait-list control group will be offered the opportunity to receive ABM+real tDCS after the end of the follow-up. Participants will be individually randomised on a 1:1:1 ratio to

the intervention or control groups in equal numbers. The generation and implementation of the randomisation sequence will be conducted independently from the trial team through a randomisation administrator who is not involved in any recruitment or research activity related to the ICARUS study. Online randomisation software (Sealed Envelope, London, UK) will be used for this purpose. On participant enrolment, the researcher will contact the randomisation administrator, who will inform this researcher in charge of carrying out the intervention of the participant's allocation via phone,email or paper letter.

## Blinding

Double blinding is implemented only for the intervention group cohorts of the trial. The research assessor will remain blind to each participant's tDCS assignment within the two intervention conditions until the study data collection phase has been completed. This double blinding protocol will be ensured via administration of the tDCS (NeuroConn DC-STIMULATOR PLUS) using 'study mode'. This involves a five-digit numerical code unique to each patient will be inputted into the device prior to the participant's testing session, that will initialise either sham or real (active) stimulation. The tDCS administrator and participants will remain blind to tDCS stimulation type throughout the study. Set-up of the randomisation codes and programming of the tDCS device will be performed by an investigator not involved in the trial. To assess blinding success, each participant and the researcher will be asked to guess the treatment allocation at the end of the six treatment sessions and to indicate how certain they are of this guess. The study group allocation will be revealed to the participant after their follow-up assessment by the study randomisation administrator. In the event of a reported change in a participant's medication, or a new clinical diagnosis made during their study participation, the early unblinding of study condition for an intervention group participant will be permissible. The trial database will be maintained 'blind' until the point of study data analyses.

## Recruitment

The study will take place at the Institute of Psychiatry, Psychology and Neuroscience (IoPPN), King's College London (KCL), UK. Participants will be recruited from the Eating Disorders Service at the South London and Maudsley NHS Foundation Trust, from the KCL research recruitment webpage and social media account, and via posters placed on notice boards on KCL campuses. Participants who have previously taken part in research at the KCL Eating Disorders Unit and who have consented to be informed of future studies may also be contacted. The ICARUS study will also be advertised on the Beat (National Eating Disorders Association) website, callforparticipants. com and www.mqmentalhealth.org Potential participants will receive written and verbal study information and will be screened for eligibility. Eligible participants will

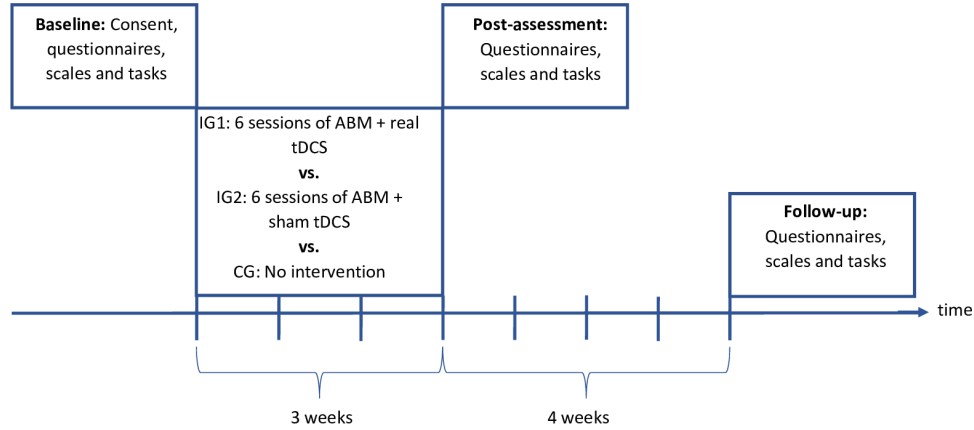

**Figure 1** Study procedure. The three assessment time points are baseline, post-assessment and follow-up. ABM, approach bias modification training; CG, wait-list control group; IG1, intervention group 1; IG2, intervention group 2; tDCS, transcranial direct current stimulation.

provide informed written consent for study participation as a prerequisite for enrolment (online supplementary appendices A and B).

## Procedure

Figure 1 illustrates the timeline of the study procedures. All participants will partake in assessments at each of the three measurement points; baseline, post-treatment and follow-up. Each assessment will comprise of an in-person study visit with tasks and measures, and online/hard-copy questionnaires and scales to be completed at home by the participant within 36 hours following the study visit. Table 1 details the tasks and measures allocated to each assessment and training visit. After the baseline assessment, participants are randomised to one of three groups: (1) ABM+real tDCS, (2) ABM+sham tDCS or a wait-list control group. Participants allocated to an intervention group will be offered six sessions of ABM+real/sham tDCS across 3 weeks. All study participants may receive TAU, for example, if they are currently engaged in outpatient treatment for their ED. The control group will not receive any study intervention, and any participants receiving outpatient services treatment for their ED will continue TAU during this 3 week period. The post-treatment assessment will be conducted on all participants after the sixth (final) session of ABM+real/sham tDCS for the intervention groups, and 3 weeks after the baseline assessment for the control group. The follow-up assessment will be conducted 28 days after the end of treatment, that is, 7 weeks post-randomisation. A follow-up period is included because if the effects of the intervention result from increased neuroplasticity, behavioural changes may need time to emerge. Assessing the longevity of favourable clinical outcomes beyond the treatment period is also relevant to the objectives of this feasibility study. The researcher conducting the assessment and testing sessions will remain blind to the study condition of intervention group participants until the study data collection period is completed.

## Outcome assessment

Measures of feasibility, safety and adherence will be collected throughout the study. Outcomes related to ED symptoms, general psychopathology and neurocognition will be measured before and after the study intervention to assess treatment success. Each assessment session will be split into an in-person visit and at-home component to accommodate time constraints and minimise disruption to task performance due to participant fatigue. The in-person assessment measures will take between 75 - 150 minutes to complete, and the online questionnaires will take approximately 30 minutes.

## Outcome measures
### Feasibility outcomes

As this is a feasibility study, an extensive range of outcome measures are included to help determine which are most sensitive to detecting a treatment effect. This will enable us to determine primary outcome(s) for a future large-scale RCT. However, based on previous research,[49] the Eating Disorder Examination Questionnaire (EDE-Q) is anticipated to be a key outcome measure.

### Intervention/service related outcomes

Feasibility outcomes include recruitment, attendance and retention rates, and acceptability of treatment by participants. Patients' acceptance of study interventions will be assessed by measuring treatment dropout rates and via the treatment tolerance and acceptability questionnaires. An interview assessment of treatment experience will be conducted with 20 participants after the follow-up is completed. Ten participants from each intervention group will be invited to provide feedback on their initial expectations and experiences of the ABM+real/sham tDCS treatments, perceived strengths and weaknesses of the treatment they received, and suggestions for improvements in procedures. Interviews will be recorded, transcribed and analysed using thematic analysis. This will allow future studies to consider patients' feedback in the

**Table 1** Study schedule of measurement and testing time points

| Approximate time since baseline | Screening of potential participants | Baseline assessment (all participants) | Training: six sessions of ABM+real tDCS 0–3 weeks | Training: six sessions of ABM+sham tDCS 0–3 weeks | Post-assessment (all participants) 3 weeks | Follow-up (all participants) 7 weeks |
|---|---|---|---|---|---|---|
| Informed consent | X | | | | | |
| EDDS, SCID-I | X | | | | | |
| tDCS safety screening | X | | | | | |
| Demographic information | | X | | | | |
| EDE-Q[83] | | X | | | X | X |
| Inhibitory control tasks; Go/No-Go task[105], SST[107] | | X | | | X | X |
| Delay discounting task with money and food[103] | | X | | | X | X |
| Food-related tasks; food choice task[88], FCT, bogus taste test[102] | | X | | | X | |
| Approach bias assessment tasks; F-AAT[85], SRC[87] | | X | | | X | X |
| Questionnaires and scales (incl. at home); EDRSQ,[119] FCQ-T-r,[94] PFS,[95] YFAS 2.0,[96] ERQ,[108] PANAS,[109] BIS-11,[120] CIA,[121] DGI,[104] DASS-21[110] | | X | | | X | X |
| Pre-(tDCS+ABM) measures: ▲ Multiple VAS, blood pressure, pulse | | | X | X | | |
| Real or sham tDCS to dLPFC | | | X | X | | |
| Approach bias modification training | | | X | X | | |
| Post-(tDCS+ABM) measures: ▲ Multiple VAS, blood pressure, pulse | | | X | X | | |
| Tolerance, discomfort and side effects | | | X | X | | |
| Acceptability questionnaire | | | | | | X |
| Blinding assessment questionnaire | | | | | | X |

ABM, approach bias modification training; BIS-11, Barrett Impulsiveness Scale; CIA, Clinical Impairment Assessment; DASS-21, Depression, Anxiety and Stress Scale; DGI, Delayed Gratification Inventory; dLPFC, dorsolateral prefrontal cortex; EDDS, Eating Disorder Diagnostic Screen; EDE-Q, Eating Disorder Examination; EDRSQ, Eating Disorder Recovery Self Efficacy Questionnaire; ERQ, Emotion Regulation Questionnaire; F-AAT, Food Approach-Avoidance Task; FCQ-T-r, Food Cravings Questionnaire Trait Version–reduced; FCT, Food Challenge Task; PANAS, Positive and Negative Affect Schedule; PFS, the 21-item Power of Food Scale; SCID-I, Structured Clinical Interview for Diagnostic and Statistical Manual (DSM) Axis I Disorders; SRC, Stimulus Response Compatibility Task; SST, Stop Signal Task; tDCS, transcranial direct current stimulation; VAS, Visual Analogue Scale; YFAS 2.0, Yale Food Addiction Scale Version 2.0.

development of research and clinical protocols of concurrent ABM and tDCS.

## Clinical outcomes
### *ED and related psychopathology*

#### Eating Disorder Examination Questionnaire
EDE-Q[83] is a widely used measure of eating-disordered behaviour and is widely regarded as the instrument of choice for the assessment of EDs. This will be administered at baseline, post-assessment and follow-up.

#### Body mass index (kg/m$^2$)
This assessment of body composition provides accurate estimates of body fat percentages in adults, where sex and age are factored into the analysis measuring height and weight.[84] To calculate BMI, height and weight measurements will be obtained by the researcher at baseline, post-assessment and follow-up as part of the EDE-Q.

#### Approach bias assessment tasks
To identify the most sensitive method of assessing change in approach bias towards high-calorie food items, two different computerised measures of approach bias will be used. In the Food Approach-Avoidance Task (F-AAT)[85] participants are shown colour photographs of high-calorie, palatable foods such as chocolate, cake and pizza, and non-food household and office items such as sponges and stationary on a computer screen.[86] They are instructed to approach and avoid these stimuli by moving a joystick toward themselves (approach) or away from themselves (avoidance). In the Stimulus Response Compatibility Task,[87] participants perform a symbolic movement by making a manikin image walk toward (approach) or away from stimuli (avoidance). See online supplementary appendix C for more detailed information. Both of these tasks will be administered at baseline, post-assessment and follow-up.

#### Food choice attitudes/behaviour
The Food Choice Task[88–90] is a computer-based paradigm that measures responses to images of foods to assess food attitudes and characteristics of eating behaviour. Participants rate images of food on a computer screen according to healthiness as well as tastiness. Based on these ratings they are then offered a choice between a food that they consider 'neutral' and a series of other foods. See online supplementary appendix C for more detailed information. This task will be performed at baseline and post-assessment.

#### Food craving after cue exposure task
The Food Challenge Task[41] will be used to examine cue-induced food craving. In this task, participants rate their state food craving using the Food Cravings Questionnaire State Version[91 92] before and after being presented with a video on a computer screen of foods shown to be highly appetising.[93] See online supplementary appendix C for more detailed information. This task will be performed at baseline and post-assessment.

#### Trait food craving
Three questionnaires will be used to comprehensively assess mechanisms implicated in trait food craving. The Food Cravings Questionnaire Trait Version–reduced[94] is a 15 items only reduced version of a self-report questionnaire that measures trait levels of craving for food. The 21-item Power of Food Scale[95] scale assesses the psychological influence of the mere presence or availability of food. It measures appetite for, rather than consumption of, palatable foods, at three levels of food proximity (food available, food present and food tasted). The Yale Food Addiction Scale Version 2.0[96] reflects the current diagnostic understanding of addiction to further investigate the potential role of an addictive process in problematic eating behaviour.[75 97–101] See online supplementary appendix C for more detailed information. Each of these scales will be administered at baseline, post-assessment and follow-up.

#### Food intake in a bogus taste test
During the bogus taste test,[102] participants will be instructed to rate and optionally consume highly palatable high-calorie food items presented in three bowls. See online supplementary appendix C for more detailed information. This task will be performed at baseline and post-assessment.

#### Preference for immediate versus delayed rewards
The Delay Discounting Task with Money and Food[103] examines whether small amounts of food would be discounted more steeply than money, as occurs with larger amounts. See online supplementary appendix C for more detailed information. The Delayed Gratification Inventory assesses participants' ability/tendencies to delay gratification for five domains (food, physical pleasures, social interactions, money and achievement[104]). Both task and questionnaire will be administered at baseline, post-assessment and follow-up.

#### Inhibitory control
The cued Go/No-Go computer task is a classic test of executive function, requiring effortful response inhibition and measures impulse control by the ability to inhibit instigated, prepotent responses. A food specific go/no-go task[105] measures impulsivity and response inhibition with respect to food and non-food items. The Stop Signal Task (SST)[106] measures inhibitory control. Participants are required to engage in a computer task but withhold their response in the presence of a stop signal. An adaptation of the food version of the SST[107] will facilitate a comparison of responses between food and non-food categories. See online supplementary appendix C for more detailed information. Both of these tasks will be performed at baseline, post-assessment and follow-up.

### Mood and emotion regulation

The Emotion Regulation Questionnaire[108] is designed to measure respondents' tendency to regulate their emotions regarding cognitive reappraisal and expressive suppression. The Positive and Negative Affect Schedule[109] measures the degree of positive or negative affect experienced "right now" in the current study. The Depression, Anxiety and Stress Scale[110] evaluates mood, anxiety and stress levels over the previous week. See online supplementary appendix C for more detailed information. All of these measures will be administered at baseline, post-assessment and follow-up.

### Within session measures

Within each training session, that is, immediately before and after the ABM+real/sham tDCS procedure the researcher will administer paper-based VAS assessing current hunger, feeling of fullness, urge to eat, urge to binge eat, feeling low, level of tension, level of stress, level of anxiety and any discomfort due to tDCS and ABM in the training session. See online supplementary appendix C for more detailed information.

### Intervention

In both intervention groups, participants will receive six sessions of concurrent ABM and real or sham tDCS which will be delivered twice a week for 3 weeks. A researcher trained in tDCS administration will deliver the training sessions.

### Rationale for number of sessions

Treatment parameters for interventions of ABM and tDCS separately in psychiatric disorder research have not yet been standardised and vary from 1 to 12 sessions across a timeframe of days to multiple weeks. Mixed results regarding optimal frequency of ABM sessions and related forms of cognitive bias modification has been reported.[111] A maximum accumulative effect of modification efficacy at six sessions has been found for ABM for alcohol dependence.[44] While there is a similar paucity of specifications for treatment parameters within tDCS, multi-session NIBS interventions are significantly more effective at reducing cravings and strengthening the ability to refrain from food consumption than single-session protocols in EDs and obesity.[112] As a single session of tDCS on patients with BED was found to reduce craving and caloric intake,[59] it was hypothesised that repeated administration of tDCS would enhance this effect and may decrease binge eating frequency.

### Within session safety procedures

The participant's blood pressure and heart rate will be taken by the researcher immediately before and after the session. While the participant is comfortably seated, the tDCS and ABM will be administered at the same time, that is, participants will engage in ABM training whilst receiving tDCS. Each session will last 20 min. The ABM training will start 5 min after the start of the brain stimulation. ABM training will take place over 10 min and tDCS will then continue for a further 5 min. Participants will be reminded that they have the option to withdraw immediately and terminate their participation in the study if they experience discomfort during tDCS administration, or if they wish to withdraw for any reason that they may or may not wish to disclose.

### Approach bias modification training

The ABM programme will use an implicit learning paradigm, based on a modified version of the F-AAT.[85 113–115] In this task, participants are shown computer images of food and control (ie, neutral office) items. They are required to pull (pictures grow bigger) or push (pictures grow smaller) a joystick in response to the outer frame of the picture (round vs rectangular), irrespective of the picture content. The training version of the Food-AAT utilises an implicit learning paradigm by presenting all food pictures in the 'push' (ie, avoid) format. The study procedure for ABM administration is aligned with previous research.[50 116]

### Transcranial direct current stimulation

tDCS (both real and sham) will be delivered using a NeuroConn DC-STIMULATOR PLUS device at a constant current of 2 mA (with a 10 s fade in/out) using two 25 cm² surface sponge electrodes soaked in a sterile saline solution (0.9% sodium chloride). The anode will be placed over the right dlPFC and the cathode over the left dlPFC. This montage has been used in sham-controlled studies on food craving, BN and BED.[59] The stimulation site will be calculated to correspond to the F3 location, as based on the International 10–20 system. tDCS can occasionally result in mild discomfort during administration (ie, tingling or itching sensation, a slightly metallic taste, occasional redness at the site of the electrodes). Fatigue, headache, nausea and insomnia have been reported as potential adverse reactions.[117] Participants who are at-risk for adverse effects[76] will be excluded from the study at the screening stage.

### Data analysis

Data will be analysed with the Statistical Package of Social Sciences (SPSS) version 26. Feasibility outcome data will be analysed with appropriate summary statistics. To determine quality, completeness and variability of the clinical outcome data, descriptive statistical analyses and graphical methods will be used. Intent-to-treat analyses will be performed. The size of the treatment effect on each outcome measure will be the difference in outcome data between those in the two treatment conditions and control condition at post-assessment and follow-up. Group differences will be estimated using linear mixed effects regression models, controlling for the baseline level of the outcome. The goal here is not to determine significant group differences but to establish a suitably precise effect size for the primary outcome at the post treatment assessment. This estimate will be used to guide the sample size of a future efficacy trial. Correlational analyses may

be computed to analyse relationships between outcome variables and influences of potential covariates such as demographic variables (ie, gender, age, BMI and clinical variables), that is, start/stopping of psychotherapy, psychotropic medication and presence of comorbidities. Outcome data already obtained for participants who discontinue or deviate from the intervention protocol will be kept and analysed.

## Patient and public involvement

Patients and/or public were not involved in the study design process, however we will obtain 20 intervention participants' qualitative views on their treatment experience in this study to inform future clinical trials.

# ETHICS AND DISSEMINATION
## Data management and data monitoring

Participant data will be anonymised and all anonymised data will be stored electronically on a password protected computer at the IoPPN. All trial data will be stored in line with the General Data Protection Regulation 2018. Hard copies of participant-related data (ie, General Practitioner letters) will be kept in locked cabinets at the IoPPN, KCL. The final trial data set will not be accessed by anyone other than members of the research team.

Data will be stored on manual files, university and laptop computers. There will be no personal data stored on laptop computers. Confidentiality and anonymity of all personal data will be retained throughout the entire study. Manual files will be securely locked in a lockable filing cabinet, and all electronic files will be password protected. Identifying information will be removed from the data, stored separately and replaced with a numeric identification code. All participants will be allocated a numeric code, which will be used to identify their data. The master list of names which correspond to each participant's numeric identification code will be stored electronically and will be password protected. This information will only be accessible to key researchers involved in the study.

The online component of the assessment will use Online Surveys software (formerly BOS). KCL uses this software for large scale surveys, and it is fully compliant with UK data protection laws. Participants will be emailed the link after the in-person component of each assessment session, and instructed to complete the second online component of the assessment within 36 hours. Participants will also have the option to receive and complete a hard copy version of the questionnaires with a stamped addressed envelope to post back to the study researcher at the IoPPN.

It is intended that the results of this feasibility study will be reported and disseminated at national and international conferences. Research findings may also be disseminated through internal newsletters and publications in collaboration with Beat, the UK's largest ED charity.

Owing to the size and nature of this small-scale feasibility study, a data monitoring committee was not deemed to be required. There are no scheduled interim analyses and this trial may be prematurely discontinued by the chief investigator on the basis of new safety information.

## Ethics and safety aspects

This trial will be conducted in compliance with the study protocol, the Declaration of Helsinki, the principles of good clinical practice (ICH-E6 guideline), the ICH-E8 guideline and the principles of good clinical practice and in accordance with all applicable regulatory requirements including but not limited to the UK policy framework for health and social care research. All participants will be asked by the study researcher to provide written informed consent prior to enrolment. Participants who are at-risk for adverse side effects[76] will be excluded from the study at the screening stage (ie, such as those with pregnancy or epilepsy). Current safety parameters of tDCS administration regarding voltage amplitude and duration of brain stimulation sessions will be adhered to. Participants have the option to withdraw immediately and terminate their participation in the study if they experience discomfort during tDCS administration, or if they wish to terminate their participation for any other reason that they may or may not wish to disclose. After each training session, participants will complete the tolerance, discomfort and side effects questionnaire to report any adverse effects of the intervention training session. The researcher will record this description of any reported adverse effects, and record the severity and duration of symptoms and how the adverse effect was managed at the following training session. If a participant reports a new clinical diagnosis or change in medication during their involvement in the study, a decision regarding their continued participation in the study will be made by the research team and withdrawal of the participant may be deemed necessary. Standard KCL insurance and NHS indemnity arrangements apply to this study. To promote study adherence, on completion of the follow-up assessment, each participant will be reimbursed for their time, efforts and travel (£60 for assessments and up to £60 for travel expenses). Additionally, participants in the wait-list control group will be offered the opportunity to receive 6 sessions of ABM+tDCS after the follow-up.

# DISCUSSION

The ICARUS study represents the first feasibility study that aims to exploit a synergistic therapeutic effect by combining two brain-directed interventions in a single treatment intervention for BED. The rising clinical need of individuals with BED is currently met with few available psychological and neuropharmacological treatment options.[4] Therefore, such research advancing the identification and validation of novel therapies is greatly warranted.

This paper delineates the protocol for a feasibility trial which will inform future studies (ie, provide effect sizes for a large RCT) and contribute to the extant research advocating brain-directed interventions for BED. The protocol aligns with current parameters of tDCS administration used to treat BED and BN[58 59] and utilises a multitudes of measures to identify the most appropriate and sensitive tools to detect treatment induced changes across pathological and neurocognitive domains.

Pragmatic concerns related to the recruitment process entail ensuring a sufficient and consistent rate of participant enrolment to meet the target sample number within the allocated timeframe. Additionally, drop-out rates for CBT treatment among a BED cohort are moderately high (17%–30%),[118] thus study adherence will need to be monitored, with a revision of incentives/ reimbursement if necessary. Participants who were randomly allocated to the wait list control may avail themselves of six sessions of ABM+real tDCS after they have completed the study, which may promote recruitment and participant retention. Documenting the management of these issues will help to inform the development of a future large-scale RCT of this combined treatment adjunct for BED.

To conclude, investigating novel treatments for BED is an imperative issue. Combining ABM with tDCS is the strategic amalgamation of two techniques that have already demonstrated therapeutic efficacy in their own right. This feasibility RCT will be the first to systematically assess the acceptability and efficacy of a non-invasive, safe and potentially effective treatment adjunct to other therapies which will enhance the ability of healthcare services to provide optimal care to patients with BED.

### Trial progress

Recruitment commenced in March 2019 and data collection is expected to be complete (including follow-up assessments) by June 2020. Any substantial protocol amendments will be communicated to investigators via email and to other parties as required. Amendments to the study protocol will be reported in publications reporting the study outcomes.

**Contributors** GG, IC, US and TB were each involved in the study conception and design. GG drafted the study protocol and IC, US and TB reviewed the paper and informed subsequent drafts. GG, IC, US and TB approved the final protocol paper.

**Funding** The project is supported by the NIHR Biomedical Research Centre (BRC) at South London and Maudsley NHS Foundation Trust (SLaM) and King's College London (KCL). Gemma Gordon is supported by a BRC PhD studentship and Ulrike Schmidt is supported by an NIHR Senior Investigator Award and receives salary support from the NIHR Mental Health BRC at SLaM NHS Foundation Trust and KCL.

**Disclaimer** The views expressed are those of the author(s) and not necessarily those of the NHS, the NIHR or the Department of Health.

**Competing interests** None declared.

**Patient consent for publication** Not required.

**Ethics approval** Ethical approval was given by the North West - Liverpool East Research Ethics Committee (ref: 18/NW/0648).

**Provenance and peer review** Not commissioned; externally peer reviewed.

**ORCID iD**
Gemma Gordon http://orcid.org/0000-0003-3888-0202

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
