## [Reviewer comments · BMJ Open]

ARTICLE DETAILS

TITLE (PROVISIONAL)	Combining cognitive bias modification training (CBM) and transcranial direct current stimulation (tDCS) to treat binge eating disorder: study protocol of a randomised controlled feasibility trial
AUTHORS	Gordon, Gemma; Brockmeyer, Timo; Schmidt, Ulrike; Campbell, Iain

VERSION 1 – REVIEW

REVIEWER	Phillipa Hay Western Sydney University Australia
REVIEW RETURNED	01-Apr-2019

GENERAL COMMENTS	The present study is a protocol of a randomised controlled feasibility (not efficacy) study of two novel treatments for binge eating disorder - transcranial Direct Current Stimulation (tDCS) and a cognitive therapy Approach Bias Modification training (ABM). The target group is people with BED who are overweight or higher BMI (BMI>250). The protocol follows quality methods for an RCT with allocation concealment, external randomisation and blinding to group were possible. The theoretical rationale is well explained and the interventions are very novel. Outcome measures use validated instruments and experimental tasks. It is an ambitious trial and either intervention would merit a trial in itself. Testing both does make this a very complex trial with a large number of outcome to be assessed. Piloting is important here. I have the following queries: Technically this is a pilot of a RCT -a subset of feasibility? see Eldridge SM, Lancaster GA, Campbell MJ, Thabane L, Hopewell S, Coleman CL, et al. (2016) Defining Feasibility and Pilot Studies in Preparation for Randomised Controlled Trials: Development of a Conceptual Framework. PLoS ONE 11(3). There also appears to be an overemphasis on assessing efficacy and determining sample size rather than piloting methods – it is recommended that participants are also advised it is a pilot or feasibility study? (see Arain et al. BMC Med Res Methodol. 2010 Jul 16;10:67. doi: 10.1186/1471-2288-10-67.
---

REVIEWER	Kathrin Schag University Hospital Tübingen Department of Psychosomatics and Psychotherapy Tübingen, Germany
REVIEW RETURNED	06-May-2019

GENERAL COMMENTS	Review of a study protocol concerning a cognitive training combined with neurostimulation in patients with BED
--

	This RCT about the feasibility and efficacy of a cognitive training program in combination with tDCS is a highly relevant topic in patients with BED as treatment of BED needs to be ameliorated. The introduction is written clearly and the research questions are derived precisely. The methodology is explained straight forward. My main issue is about the clarification of how treatment as usual (TAU) is included in the trial: At the study aims section, it is written that the training program is an “adjunct to TAU”, whereas in the abstract and introduction, it is not mentioned that the patients do TAU also. Further, at the Study design section, it is mentioned that the patients “will continue” TAU which indicates that the patients do not have TAU parallel to the training, but take a break of TAU while the training is running. At the procedure however, it is written that patients in the intervention groups “may also receive TAU” and that the control group “may continue TAU” during the intervention period. I recommend to review these sections and write more clearly, also in the abstract and introduction of how and when patients are allowed to take part on TAU. The rationale for this should also be justified. Moreover, additional TAU will produce treatment effects that are not due to the cognitive training program with or without tDCS. Thus, it should be clarified how the study team will handle such treatment effects, particularly if there is only a subgroup of patients with TAU, if I understand right. Will those who receive TAU be balanced across the conditions or will TAU effects be controlled statistically? Please write more precisely in the data analysis section how you will deal with such effects. Another point is that I was missing which concrete differences between the ABM + sham tDCS vs. the waitlist control you would expect. In my opinion, the training program alone (sham tDCS) should also be superior to the waitlist control as it should elicit training effects. The tDCS might serve as a booster of the training and thus be superior to the training program alone. This should be included at the study aim section. Next, at the supplemental (participant information sheet) it is written that there will be a 6 months telephone follow-up. Why is this follow-up not included into the study protocol? Please add this to the design of the study and to figure 1. Further, it should be included in the abstract that a “waitlist” control group is used. At the tDCS section, add with how much mAmp you will stimulate. Also, please clarify in the data analysis section which factors you will include into your analysis. The legend of table 1 should be sorted alphabetically. Last, I miss the year from reference 65.
--	---

VERSION 1 – AUTHOR RESPONSE

Reviewer #1

The present study is a protocol of a randomised controlled feasibility (not efficacy) study of two novel treatments for binge eating disorder - transcranial Direct Current Stimulation (tDCS) and a cognitive therapy Approach Bias Modification training (ABM). The target group is people with BED who are overweight or higher BMI (BMI>25).

The protocol follows quality methods for an RCT with allocation concealment, external randomisation and blinding to group were possible. The theoretical rationale is well explained and the interventions are very novel. Outcome measures use validated instruments and experimental tasks. It is an

ambitious trial and either intervention would merit a trial in itself. Testing both does make this a very complex trial with a large number of outcome to be assessed. Piloting is important here.

I have the following queries:

1. Technically this is a pilot of a RCT -a subset of feasibility? see Eldridge SM, Lancaster GA, Campbell MJ, Thabane L, Hopewell S, Coleman CL, et al. (2016) Defining Feasibility and Pilot Studies in Preparation for Randomised Controlled Trials: Development of a Conceptual Framework. PLoS ONE 11(3).

Thank you for highlighting the recent developments that have entailed a review of the definitions of, and differences between, feasibility and pilot studies. The present research undertaken is fully funded by a National Institute of Health Research (NIHR) PhD studentship, and at the time of the study's design in early 2018, this study's research followed the guidance issued by the NIHR regarding the use of these terms. To act on your helpful comment, we have contacted the NIHR to establish if the current classification of the present study as a feasibility study remains most representative. The response was that the paper "NIHR guidance on feasibility studies" best represents the NIHR position so far and thus we have decided to remain with the current description. To clarify this, I have added an explanatory sentence to the Study Aims paragraph on page 9:

In line with established recommendations for outcomes of feasibility trials[72], which at present are supported by the National Institute for Health Research, the primary aim is to assess the feasibility of using concurrent ABM + real tDCS compared to concurrent ABM + sham tDCS as a potential adjunct to treatment as usual (TAU) in this patient population, and acquire key information to inform the development of a large-scale randomised sham-controlled trial (RCT).

2. There also appears to be an overemphasis on assessing efficacy and determining sample size rather than piloting methods – it is recommended that participants are also advised it is a pilot or feasibility study? (see Arain et al. BMC Med Res Methodol. 2010 Jul 16;10:67. doi: 10.1186/1471-2288-10-67).

Thank you for this comment and recommendation of relevant literature. The primary study aims outlined in this protocol paper are closely aligned with those of feasibility studies presented in the Arain et al. (2010) paper, under the NETSCC definition of pilot and feasibility studies. It has been noted that participants could be better informed about whether this is a pilot or feasibility study prior to their participation. As such, an amendment to the participant information sheet will be made to clarify this.

Reviewer # 2

Review of a study protocol concerning a cognitive training combined with neurostimulation in patients with BED

This RCT about the feasibility and efficacy of a cognitive training program in combination with tDCS is a highly relevant topic in patients with BED as treatment of BED needs to be ameliorated. The introduction is written clearly and the research questions are derived precisely. The methodology is explained straight forward.

3. My main issue is about the clarification of how treatment as usual (TAU) is included in the trial: At the study aims section, it is written that the training program is an "adjunct to TAU", whereas in the abstract and introduction, it is not mentioned that the patients do TAU also. Further, at the Study design section, it is mentioned that the patients "will continue" TAU which indicates that the patients do not have TAU parallel to the training, but take a break of TAU while the training is running. At the procedure however, it is written that patients in the intervention groups "may also receive TAU" and that the control group "may continue TAU" during the intervention period. I recommend to review these sections and write more clearly, also in the abstract and introduction of how and when patients are allowed to take part on TAU. The rationale for this should also be justified.

Thank you for highlighting this. The listed sections (with the exception of the Abstract due to word limitations) have been edited to present a clearer and more consistent summary of the nature and

potential presence of TAU among participants. The following clarification has been incorporated into the introduction, page 9:

“In light of both the individual and societal burden incurred by the rising prevalence of BED and obesity, research interventions informing treatments that lead to stable and long-lasting remission are of critical importance, and novel therapies may play a role in serving as adjuncts to treatment as usual, to enhance improvement in clinical outcomes obtained from engaging with eating disorder treatment services.”

4. Moreover, additional TAU will produce treatment effects that are not due to the cognitive training program with or without tDCS. Thus, it should be clarified how the study team will handle such treatment effects, particularly if there is only a subgroup of patients with TAU, if I understand right. Will those who receive TAU be balanced across the conditions or will TAU effects be controlled statistically? Please write more precisely in the data analysis section how you will deal with such effects.

Thank you for raising this point. A discussion of treatments constituting TAU has now been included in the data analysis section on page 16, which will address any treatment effects from TAU. As a proportion of participants in this study will be recruited from outpatient eating disorder services where they may be on a waiting list for treatment at the time of study enrolment, they may commence treatment shortly afterwards which would disrupt attempts to counterbalance participants across condition. As such, at each study visit (for all assessments and treatment administration sessions), participants are instructed to detail in writing if they have started or stopped any treatment for their eating disorder symptoms, and the details of such treatment (i.e. psychotherapy vs. psychotropic medication). TAU effects are monitored and will be controlled for via covariate analyses.

Correlational analyses may be computed to analyse relationships between outcome variables and influences of potential covariates such as demographic variables, i.e. gender, age, BMI and clinical variables, i.e. start/stopping of psychotherapy, psychotropic medication and presence of comorbidities. Outcome data already obtained for participants who discontinue or deviate from the intervention protocol will be kept and analysed.

5. Another point is that I was missing which concrete differences between the ABM + sham tDCS vs. the waitlist control you would expect. In my opinion, the training program alone (sham tDCS) should also be superior to the waitlist control as it should elicit training effects. The tDCS might serve as a booster of the training and thus be superior to the training program alone. This should be included at the study aim section.

Thank you for this comment. The Study Aims section continued on page 10 now includes a third point under the secondary aims which outlines that this study will evaluate if the ABM + sham tDCS intervention group yields therapeutic effects in participants to a greater degree than the waitlist control group, but to a lesser degree than the ABM + real tDCS group.

A secondary aim is to investigate the potential efficacy of concurrent delivery of both forms of treatment on binge eating disorder.

This will involve evaluating if:

1. Concurrent sessions of ABM + real tDCS are superior to ABM + sham tDCS and to wait-list control in terms of frequency of objective binge eating episodes, food cue reactivity, food craving, food intake, eating disorder psychopathology and mood.

2. Concurrent ABM + tDCS is superior to the two other conditions in having an effect on the targeted neurocognitive mechanism (approach bias for high calorie food) and related neurocognitive parameters (i.e. impulsivity, delayed gratification, emotional regulation).

3. Concurrent ABM and sham tDCS is superior to the waitlist control in eliciting therapeutic effects on the aforementioned clinical outcomes and neurocognitive mechanisms, yet demonstrates an efficacy level below that of concurrent ABM and real tDCS.

6. Next, at the supplemental (participant information sheet) it is written that there will be a 6 months telephone follow-up. Why is this follow-up not included into the study protocol? Please add this to the design of the study and to figure 1.

In the participant information sheet, the 6 month telephone follow-up phone call is presented as a possibly occurrence (“participants may be contacted”), to notify potential participants that this may happen, although at present their participation in the study is officially complete upon the 1-month follow up assessment (it is stated that they are reimbursed for their participation after this 1-month follow up assessment). This 6 month phone call was not part of the study protocol submitted and approved by ethics, and is a polite notice to indicate to potential study participants that there may be a protocol amendment undertaken at a future time to facilitate this event. Due to the temporal and financial constraints of this research study, which is funded by a PhD studentship grant solely allocated to the main researcher, data analyses will begin after all participants have completed the 1-month follow-up assessment. In the event of this 6-month phone call being implemented, any data collected will not constitute part of this study’s main analysis, and would be part of a wider dataset obtained by the research department monitoring any long-term effects of their ongoing research studies administering non-invasive brain stimulation interventions.

7. Further, it should be included in the abstract that a “waitlist” control group is used.

Thank you, this word has now been incorporated into the abstract.

8. At the tDCS section, add with how much mA you will stimulate.

The text on page 16 had stated that a current of 2 mA would be used “TDCS (both real and sham) will be delivered using a NeuroConn® DC-STIMULATOR PLUS device at a constant current of 2 mA (with a 10-second fade in/out) using two 25cm² surface sponge electrodes soaked in a sterile saline solution (0.9% sodium chloride).” This presentation format of current strength is consistent with literature on tDCS (i.e. Kekic, M., McClelland, J., Bartholdy, S., Boysen, E., Musiat, P., Dalton, B., ... & Schmidt, U. (2017). Single-session transcranial direct current stimulation temporarily improves symptoms, mood, and self-regulatory control in bulimia nervosa: a randomised controlled trial. *PLoS one*, 12(1), e0167606.; den Uyl, T. E., Gladwin, T. E., Lindenmeyer, J., & Wiers, R. W. (2018). A Clinical Trial with Combined Transcranial Direct Current Stimulation and Attentional Bias Modification in Alcohol-Dependent Patients. *Alcoholism: Clinical and Experimental Research*, 42(10), 1961-1969).

9. Also, please clarify in the data analysis section which factors you will include into your analysis.

I have sought to address this point while acting upon point 4. Please find the factors specified in more detail in the data analysis section on page 16.

10. The legend of table 1 should be sorted alphabetically.

Thank you, the legend has been amended accordingly on page 30.

11. Last, I miss the year from reference 65.

Thank you for highlighting this error, the formatting of this citation on page 25 has been rectified.

VERSION 2 – REVIEW

REVIEWER	Phillipa Hay Western Sydney University Australia
REVIEW RETURNED	29-Jul-2019

GENERAL COMMENTS	Most comments have been addressed. I have one query re: "Total sample sizes of n=24 to n=50 have been recommended for feasibility trials with a primary outcome measured on a continuous scale, ..." Why not have 24 per group or 72 which is the lower limit of the cited recommended number? Other points e.g. balancing groups for medication use and randomisation will be important for the later RCT if this feasibility trial is successful.
---

REVIEWER	Kathrin Schag
-----------------	---------------

	University Hospital Tuebingen, Psychosomatic Medicine and Psychotherapy, Tuebingen, Germany
REVIEW RETURNED	19-Jul-2019

GENERAL COMMENTS	The authors addressed all my questions. I have nothing to add.
--

VERSION 2 – AUTHOR RESPONSE

Responses to Reviewers

Reviewer: 2

The authors addressed all my questions. I have nothing to add.

Reviewer 1

Most comments have been addressed.

- I have one query re:"Total sample sizes of n=24 to n=50 have been recommended for feasibility trials with a primary outcome measured on a continuous scale, ..."Why not have 24 per group or 72 which is the lower limit of the cited recommended number?

Our understanding of the cited article is that it is a recommendation for a total study sample size for feasibility studies. In our study, we will have a total sample size of n=66, ie the upper recommended number (n=50) has been exceeded. We note however, that there may be some ambiguity in our text ie in relation to whether the number cited indicates per study group or overall sample size. To make this clear, we have now inserted the word 'study' twice in the relevant paragraph on page 7. The sentence now reads:

Total study sample sizes of n=24 to n=50 have been recommended for feasibility trials with a primary outcome measured on a continuous scale, mainly because estimates of the standard deviation for normally distributed variables tend to stabilise around this size[79,80]. We have chosen a target end study sample size of n=60, (i.e. this exceeds the upper end recommended for feasibility trials).

- Other points e.g. balancing groups for medication use at randomisation will be important for the later RCT if this feasibility trial is successful.

We are aware that this is an important issue and it is one that we have considered. We have not sought to balance the groups in the present study for the following reasons. As a proportion of participants in this feasibility study are recruited from an outpatient clinic where they have recently been assessed and are awaiting treatment for their eating disorder, usage of medication for an eating disorder or a comorbid psychiatric condition (depression and/or anxiety) may stop or start during their 7 weeks of participation in this study. As such, an attempt to balance groups according to the presence/absence of medication usage at baseline is not indicative of their circumstances throughout study participation. To accurately track the starting/stopping of therapy and medication usage, participants are asked via written questions at each assessment and intervention visit to record if there have been any changes in medication usage for any psychiatric or physiological illness diagnosis. To justify the absence of group balancing based on this rationale, the following sentence has now been inserted into the text on page 7 of the manuscript.

Randomisation

After the baseline assessment, participants will be allocated to one of three conditions at random to receive 6 sessions of either concurrent ABM + real tDCS or ABM + sham tDCS, or no intervention in the wait-list control condition. As a proportion of participants recruited from an outpatient eating disorder clinic will be on a waiting list to receive treatment at the time of enrollment in this study and may commence treatment shortly after study enrollment, this study will not seek to balance groups in terms of therapy engagement or medication usage.

We hope these changes address the points raised by the referee.

VERSION 3 – REVIEW

REVIEWER	Phillipa Hay Western Sydney University Australia
REVIEW RETURNED	02-Aug-2019

GENERAL COMMENTS	The reviewer completed the checklist but made no further comments.
--